# The impact of the newly developed school-based 'Digital Health Contact'—Evaluating a health and wellbeing screening tool for adolescents in England

**Alice Porter**[1]*, **Katrina d'Apice**[1], **Patricia Albers**[1], **Nicholas Woodrow**[2], **Hannah Fairbrother**[3], **Katie Breheny**[1], **Clare Mills**[4], **Sarah Tebbett**[5], **Frank De Vocht**[1,6]

1 Population Health Sciences, Bristol Medical School, University of Bristol, Bristol, United Kingdom, 2 School of Health and Related Research (ScHARR), University of Sheffield, Sheffield, United Kingdom, 3 Health Sciences School, University of Sheffield, Sheffield, United Kingdom, 4 Public Health, Leicester City Council, Leicester, United Kingdom, 5 Leicestershire Partnership NHS Trust, Leicester, United Kingdom, 6 NIHR Applied Research Collaboration West (NIHR ARC West), Bristol, United Kingdom

* alice.porter@bristol.ac.uk

**Data Availability Statement:** Data on schools, pupils and their characteristics are publicly available here: https://explore-education-statistics.

## Abstract

### Introduction

Supporting adolescents with their health and wellbeing is an international public health priority. Schools are well placed to universally detect unmet health needs and support pupils. This study aimed to evaluate the effectiveness of a digital health and wellbeing screening tool, called the 'Digital Health Contact' (DHC) implemented in schools in the East Midlands of England. The DHC, delivered by Public Health Nurses (School Nurses) (PHN(SN)), aims to identify pupils with unmet health needs (via a 'red flag' system) and provide appropriate support.

### Materials and methods

Using data from 22 schools which took part in the DHC and 14 schools which did not take part, across three academic years (2018–2020), we conducted a controlled interrupted time-series analysis with negative binomial regression to explore the effect of the DHC on the number of annual referrals to PHN(SN). Using DHC data from 164 pupils, we further conducted a Difference-in-Difference analysis to explore the impact of 'red flag' and referral via the DHC in Year 9 (age 13–14) on the number of red flags in Year 11 (age 15–16).

### Results

Across all schools, the mean annual number of referrals increased over the three year follow-up period. In the adjusted model, the number of referrals was comparable between schools taking part in the DHC and non-participating schools (0.15 referrals [95% CI -0.21, 0.50]). Red flag score was not significantly different among Year 11 pupils, after being referred via the DHC in Year 9 (-0.36 red flags [95% CI -0.97, 0.24]).

service.gov.uk/find-statistics/school-pupils-and-their-characteristics/2020-21 The study data includes school and pupil-level identifiable data related to the Digital Health Contact. The study data cannot be shared publicly because the authors do not have permission to share, due to the privacy and ethics restrictions set out in the data sharing agreement. The study data is owned by a third party (Leicester City Council and Leicestershire Partnership NHS Trust). The study was developed with Leicester City Council and Leicestershire Partnership NHS Trust and the research team were granted access to the study data for the purpose of the study only. The study data may be accessed by contacting Leicester City Council directly (customer.services@leicester.gov.uk).

**Funding:** This study is funded by the National Institute for Health and Care Research (NIHR) School for Public Health Research (SPHR) (Grant Reference Number SPHR-PHPES009-DHC). The grant was awarded to FDV. The NIHR School for Public Health Research is a partnership between the Universities of Sheffield, Bristol, Cambridge, Imperial, University College London, The London School for Hygiene and Tropical Medicine (LSHTM), LiLaC — a collaboration between the Universities of Liverpool and Lancaster and The Centre for Translational Research in Public Health (Fuse) a collaboration between Newcastle, Durham, Northumbria, Sunderland and Teesside Universities. The views expressed are those of the author(s) and not necessarily those of the NIHR or the Department of Health and Social Care. The funders had no role in study design, data collection and analysis, decision to publish, or preparation of the manuscript.

**Competing interests:** AP, KD, PA, NW, HF, KB and FDV have no competing interests. The funders had no additional role in this study. We should note that CM is employed at the DHCs commissioning organisation (at Leicester City Council), and ST is employed at the DHCs provider organisation (at Leicestershire Partnership NHS Trust). This study, and the wider research project evaluating the DHC, has been co-produced with the DHC's commissioners (at Leicester City Council) and providers (at Leicestershire Partnership NHS Trust). The commissioner and the provider are committed to helping to facilitate a robust evaluation of the DHC in order to inform potential changes and wider roll out of the programme. There has been no pressure or influence to modify, restate, weaken, omit or frame findings, conclusions and recommendations from any team member. The study did not receive funding from the commissioner or provider.

## Discussion

The DHC, and similar screening tools, have the potential to raise awareness of the health and wellbeing support in schools and provide an additional pathway of referral to this support for pupils with unmet health needs, without replacing the traditional pathway where pupils refer themselves or are referred by teachers.

## Introduction

Globally, around a third (31%) of adolescents are estimated to suffer from a common mental health disorder [1]. In the UK, one in six (17%) children and young people (aged six to 16 years) were estimated to suffer from a mental health disorder in 2021; a 6% increase since 2017 [2]. Further evidence suggests the prevalence of mental health issues such as emotional problems, conduct problems and hyperactivity are even higher, with two in five adolescents suffering [3]. Adolescents face a range of academic, personal and social challenges during the school years. A large community-based survey in England (including 28,160 adolescents) showed that reporting of mental health issues increased from Year 7 (11–12 years) to Year 11 (15–16 years), and was higher among adolescents with Special Educational Needs (SEN) and those from deprived backgrounds. In addition, boys were more likely to report behavioural problems, whilst girls were more likely to report emotional problems [3]. Poor adolescent mental health is associated with poor educational attainment, unemployment, risky behaviours (such as substance misuse), and mental health disorders in adulthood [4–6]. The need to prioritise young people's health is therefore paramount.

Schools are well placed to monitor and introduce initiatives to improve pupil health and wellbeing [7]. A systematic review and meta-analysis indicated school-based interventions can be effective at preventing anxiety and depression, with greater impacts on reducing depression when interventions were targeted at high risk adolescents [8]. The rising prevalence of reported mental health issues suggests monitoring of adolescent mental health during secondary school is key. In addition, targeted and tailored approaches to improve adolescent mental health should be implemented and evaluated.

Adolescents can face numerous barriers to seeking mental health support, such as not knowing where to access support, perceiving their problems to not warrant support, fears around stigmatisation, and perceiving help seeking as a weakness [9]. Evidence suggests such barriers contribute to many adolescents lacking support because they may avoid seeking help from school staff, parents or professionals [10, 11]. Self-report questionnaires administered in schools could provide adolescents with the opportunity to report sensitive information in a safe environment so that they can be directed to appropriate support [12], without feeling stigmatized [8, 13, 14]. In addition, self-report questionnaires may be more effective, quicker, and less resource intensive than traditional face-to-face approaches for collecting sensitive behavioural data [15]. However, it is important school-based mental health tools are evaluated to ensure they are acceptable, effective and avoid unintended harms.

### The Digital Health Contact

The Digital Health Contact (DHC) has been described in detail elsewhere [16, 17]. It is an online, school-based, self-report, health and wellbeing screening tool, commissioned in the East Midlands of England. The DHC is completed by pupils during school. It comprises questions relating to health and wellbeing, including feelings of safety, safeguarding, mental

wellbeing, body image, physical health and substance use. The DHC also provides the option to add free text. Compared to the traditional approach, whereby pupils must refer themselves or staff must refer a pupil to their Public Health Nurse (School Nursing) (PHN(SN)), the DHC acts as a screening tool which aims to refer pupils to their PHN(SN) if they are identified as having unmet health needs based on their DHC answers. Certain answers or words in the free text responses automatically (using an algorithm) 'red-flags' a pupil to have unmet health needs. This prompts the PHN(SN) to decide whether the pupil is in need of a face-to-face consultation, known as a Baseline Health Assessment (BHA). During the BHA, the PHN(SN) will assess the pupil's mental health and provide advice and support. A PHN(SN) might offer an evidence based package of care, delivered by one of the team, or refer to more specialist support (such as NHS Child and Adolescent Mental Health Services (CAMHS), social services). S1 Fig presents the DHC flowchart, illustrating the different pathways of care. The DHC aims to capture pupils who would not necessarily refer themselves to their PHN(SN) or be identified by staff as having unmet health needs, and is intended to be used alongside the traditional approach.

This study is part of a wider mixed-methods evaluation of the DHC. The overall aim of the evaluation is to evaluate the acceptability, utility and effectiveness of the DHC in identifying and putting strategies in place to meet unmet health needs of adolescents in secondary schools. This paper reports on the evaluation of the impact on the annual number of PHN(SN) referrals and adolescent reported health and wellbeing issues.

## Methods

We describe the methods of a real-world evaluation study, including the study design, recruitment strategy, measures and statistical analysis.

### Study design

A quasi-experimental study [18] was conducted using data provided by schools. Treatment allocation was non-randomised because the DHC had already been implemented within schools in the East Midlands of England, as part of the national 0–19 Healthy Child Programme. It was therefore not possible for the researchers to control treatment allocation. Ethical approval was granted for this evaluation by The Faculty of Health Sciences Research Ethics Committee at the University of Bristol (Reference 110982).

### Recruitment

PHN(SN) were responsible for recruiting schools to take part in the DHC. Parental consent was gained via an opt-out process. Pupils could decline to complete the DHC at any time. Twenty-two schools took part in the DHC (intervention schools) across 2017/18, 2018/19, 2019/20 and 2020/21. In total, 5080 pupils across Years 7 (11–12 years), 9 (13–14 years) and 11 (15–16 years) completed the DHC. 164 pupils from one school took part in the DHC twice; in Year 9 and then again in Year 11, providing longitudinal data. Fourteen schools within the same geographical region did not take part in the DHC (they were either not approached by PHN(SN) or they declined to take part, for reasons such as logistics) but provided school-level outcome data (control schools).

### Measures

Pupil-level demographics collected via the DHC included age, gender, ethnicity and postcode. Pupil red flag score (i.e. number of red flags) and whether pupils' red flag score had

resulted in a PHN(SN) referral and appointment were collected from the DHC. Postcode was used to derive Index of Multiple Deprivation (IMD) [19], as an indicator of socioeconomic status.

School-level characteristics included school size (N pupils), percentage of girls to boys, percentage of pupils eligible for free school meals (FSM), percentage of pupils with English as their first language, percentage of pupils with Special Educational Needs (SEN) and school IMD, and were collated from publicly available UK Government datasets [20]. The annual number of referrals made to the PHN(SN) via the traditional referral pathway (via pupils or teachers) was reported by all 36 schools between 2018 and 2021. The number of referrals resulting from pupils completing the DHC was not available. The number of pupils taking part in the DHC each year was reported by intervention schools only.

## Statistical analysis

Twenty-six schools (72%) did not provide outcome data during the 2017/18 academic year, therefore data were analysed for all 36 schools (22 intervention, 14 control) across three academic years (2018/19, 2019/20, 2020/21). Descriptive statistics are presented for intervention vs control schools. We conducted a controlled interrupted timeseries analysis using negative binomial mixed effects regression models to account for repeated measures and overdispersion. An indicator value described whether a school completed the DHC in a given year. Only one school participated in all academic years, with the participation of the other 21 schools being variable across the years (and control schools not taking part at all). S1 Table shows which years each school took part. The log of school size (N of total pupils) was added as an offset variable. Model covariates included the annual number of pupils completing the DHC, whether schools had ever taken part in the DHC, percentage of girls to boys, percentage of pupils eligible for FSM, percentage of pupils with English as their first language, percentage of pupils with SEN and school IMD. To minimise measured covariate imbalances between the groups as a result of the self-selection of schools into the DHC programme, models were propensity score (PS) weighted based on 2017/18 covariate values. A doubly robust weighted model (weighted and adjusted for covariates) was used.

We conducted moderation analysis to explore whether associations between completing the DHC and annual referrals differed across different levels of the covariates. The continuous covariates were recoded into binary covariates to explore associations for schools with low vs high percentage of girls, pupils eligible vs not for FSM, pupils with English as their first language vs not, pupils with vs without SEN and for schools with below vs above median school-level deprivation. Binary covariates were added as interaction terms to the doubly robust negative binomial mixed effects models (e.g. DHC x percentage girls to boys).

Pupil-level time series data was only provided by one school. These data included DHC answers, demographics (gender, postcode) red flag score and whether a PHN(SN) referral appointment was offered, across 164 pupils in Year 9 and again in Year 11. We conducted difference-in-difference (DiD) analysis to assess whether attending a PHN(SN) referral appointment in Year 9 as a result of completing the DHC led to a reduction in red flag score over time (from Year 9 to 11), compared to pupils who did not attend a PHN(SN) referral appointment in Year 9. A negative binomial generalized linear mixed model was conducted including the main effects for attending a PHN(SN) appointment in Year 9 (i.e. attended vs did not attend) and time point (i.e. Year 9 vs Year 11), as well as the interaction effect (i.e. the effect of attending PHN(SN) appointment in Year 9 on red flag score in Year 11). The model was adjusted for number of red flags in Year 9, gender and IMD (derived from pupil-reported postcode).

Finally, to explore which patterns of the DHC screening tool items resulted in referrals, we conducted a principle component analysis (PCA) of these pupil-level data. Principle components with eigenvalues >1 were retained. All statistical analyses were conducted in Stata 15.

## Results

We presents the results of the school-level, pupil-level, and principle component analyses.

### School-level analysis

S1 Table presents school-level characteristics for all schools. Mean school size in the sample was 960 pupils (SD 388); 907 pupils (SD 412) in intervention schools and 1042 pupils (SD 412) in control schools. A total of 4031 pupils completed the DHC across the 22 intervention schools between 2018 and 2021. 1902 pupils took part in the DHC in 2018/19, 1616 in 2019/20, and 513 in 2020/21. The mean number of pupils taking part in the DHC in intervention schools was 144 (SD 100). Sixteen schools took part in the DHC in 2018/19, eight in 2019/20 and four in 2020/21 (Table 1). Across the three academic years, one intervention school took part in the DHC three times, three took part twice and 15 took part once (S1 Table).

Across all three years, the mean annual number of PHN(SN) referrals in schools taking part in the DHC was 92 (IQR 29–132) vs 80 (IQR 45–103) in schools not taking part. Table 1 shows the temporal pattern of referrals and describes an average increase over the three academic years in both groups. Referrals were higher on average in schools taking part in the DHC compared to schools not taking part in each year.

Table 2 presents the main results of the doubly robust model, as well as the basic and fully adjusted (non-weighted) models. The basic model suggested referrals were approximately 36% higher when schools took part in the DHC (0.36 referrals [95% CI 0.02, 0.72]) compared to those that did not take part. However, in the fully adjusted (0.15 referrals [95% CI -0.23, 0.52]) and doubly robust models (0.15 referrals [95% CI -0.21, 0.50]) taking part in the DHC was no longer associated with a statistically significant increased number of referrals compared to non-participating schools.

S2 Table presents the moderation analysis results, exploring potential health inequalities. There was no evidence of interactions between percentage of pupils eligible for FSM, percentage of girls, percentage of pupils with SEN, nor school IMD and numbers of referrals. An interaction (-0.55 [95% CI -1.02, -0.09]) was observed for percentage of pupils with English as their first language and increased referrals. A higher number of annual referrals (approx. 50 on average) was observed in schools taking part in the DHC vs schools not taking part in schools with a low percentage of pupils with English as their first language, which was not observed for schools with a high percentage of pupils with English as their first language.

**Table 1. Mean annual number of PHN(SN) referrals in schools taking part in the DHC vs schools not taking part, across three academic years.**

| | Schools taking part in DHC | | | Schools not taking part in DHC | | |
|---|---|---|---|---|---|---|
| | N schools | Mean number of annual PHN(SN) referrals | IQR | N schools | Mean number of annual PHN(SN) referrals | IQR |
| 2018/19 | 16 | 44 | 28–63 | 19 | 35 | 28–47 |
| 2019/20 | 8 | 152 | 117–211 | 27 | 87 | 56–116 |
| 2020/21 | 4 | 164 | 74–254 | 31 | 101 | 60–122 |

Abbreviations: DHC; Digital Health Contact, PHN(SN); Public Health Nurse (School Nursing), IQR; interquartile range

**Table 2. Basic, fully adjusted and doubly robust negative binomial mixed-effects regression models exploring the effect of the DHC on the annual number of PHN (SN) referrals (n observations = 104, n schools = 36).**

| Number of annual PHN(SN) referrals | Basic model | | | Fully adjusted model | | | Doubly robust model* | | |
|---|---|---|---|---|---|---|---|---|---|
| | Coefficient | 95% CI | p value | Coefficient | 95% CI | p value | Coefficient | 95% CI | p value |
| Treatment | | | | | | | | | |
| Taking part in the DHC | 0.36 | 0.02 0.72 | 0.040 | 0.15 | -0.23 0.52 | 0.442 | 0.15 | -0.21 0.50 | 0.416 |
| Year | | | | | | | | | |
| 2 (2019/20) | 1.02 | 0.83 1.21 | <0.001 | 0.85 | 0.66 1.04 | <0.001 | 0.89 | 0.69 1.09 | <0.001 |
| 3 (2020/21) | 1.16 | 0.96 1.35 | <0.001 | 0.84 | 0.63 1.05 | <0.001 | 0.94 | 0.69 1.20 | <0.001 |
| N pupils taking part in DHC | 0.00 | 0.00 0.00 | 0.675 | 0.00 | 0.00 0.00 | 0.110 | 0.00 | 0.00 0.00 | 0.074 |
| Ever taken part in DHC | | | | | | | | | |
| Yes | 0.20 | -0.09 0.50 | 0.181 | 0.05 | -0.13 0.24 | 0.589 | 0.07 | -0.12 0.27 | 0.471 |
| % pupils eligible for FSM | | | | 0.05 | 0.03 0.07 | <0.001 | 0.04 | 0.01 0.06 | 0.002 |
| % pupils with English as first language | | | | 0.01 | 0.00 0.01 | 0.001 | 0.01 | 0.00 0.01 | <0.001 |
| IMD | | | | 0.03 | -0.02 0.08 | 0.233 | 0.00 | -0.01 0.01 | 0.682 |
| % girls | | | | 0.00 | 0.00 0.01 | 0.462 | 0.00 | -0.05 0.04 | 0.866 |
| % pupils with SEN | | | | 0.00 | -0.02 0.02 | 0.911 | 0.00 | -0.02 0.03 | 0.826 |

Abbreviations: DHC; Digital Health Contact, PHN(SN); Public Health Nurse (School Nursing), FSM; free school meals, IMD; Index of Multiple Deprivation, SEN; Special Educational Needs

Akaike's information criterion (AIC) was calculated as a measure of model fit. In the basic model, AIC = 1021.993. In fully adjusted model, AIC = 976.2472. In the doubly robust model, AIC = 1915.896

The log of school size (total number of pupils) was included as the offset variable in all models.

*Inverse probability weights for % pupils with FSM, % pupils with English as first language, school IMD, % girls and % pupils with SEN included

## Pupil-level analysis

Fig 1 shows red flag scores in Year 9 and again in Year 11. The number of pupils with one or more red flags reduced from Year 9 (79 pupils; 48%) to Year 11 (72 pupils; 44%). Of the 79 pupils who did have at least one red flag in Year 9, 24 pupils (30%) attended a PHN(SN) appointment and 9 pupils (11%) also attended a further follow-up appointment. S3 Table presents the change in number of red flags between Year 9 and Year 11. Forty-three pupils had a lower number of red flags in Year 11 compared to Year 9, whereas 44 pupils had a higher number of red flags in Year 11 compared to Year 9. 77 pupils had the same number of red flags in Year 11 compared to Year 9 (including 67 pupils who had no red flags in Year 9 and Year 11).

Pupils who did not have any red flags in Year 9 had the lowest mean number of red flags in Year 11 (0.4 [Range 0–5]). Pupils who had one or more red flags in Year 9 and attended a PHN(SN) appointment had a higher mean number of red flags in Year 11 (2.7 [Range 0–8]) compared to those who did not attend a PHN(SN) appointment (1.6 [Range 0–6]) (Table 3). Of the 24 pupils who attended a PHN(SN) appointment in Year 9, 4 pupils had no change in the number of red flags, 8 pupils had an increase in number of red flags and 12 pupils had a decrease in number of red flags, from Year 9 to Year 11.

Table 4 presents results of the statistical modelling exploring the effect of attending a PHN (SN) referral appointment in Year 9 on red flag score in Year 11. On average, pupils in Year 11 who had attended a PHN(SN) appointment in Year 9 did not have a significantly lower red flag score (-0.36 red flags [95% CI -0.97, 0.24]).

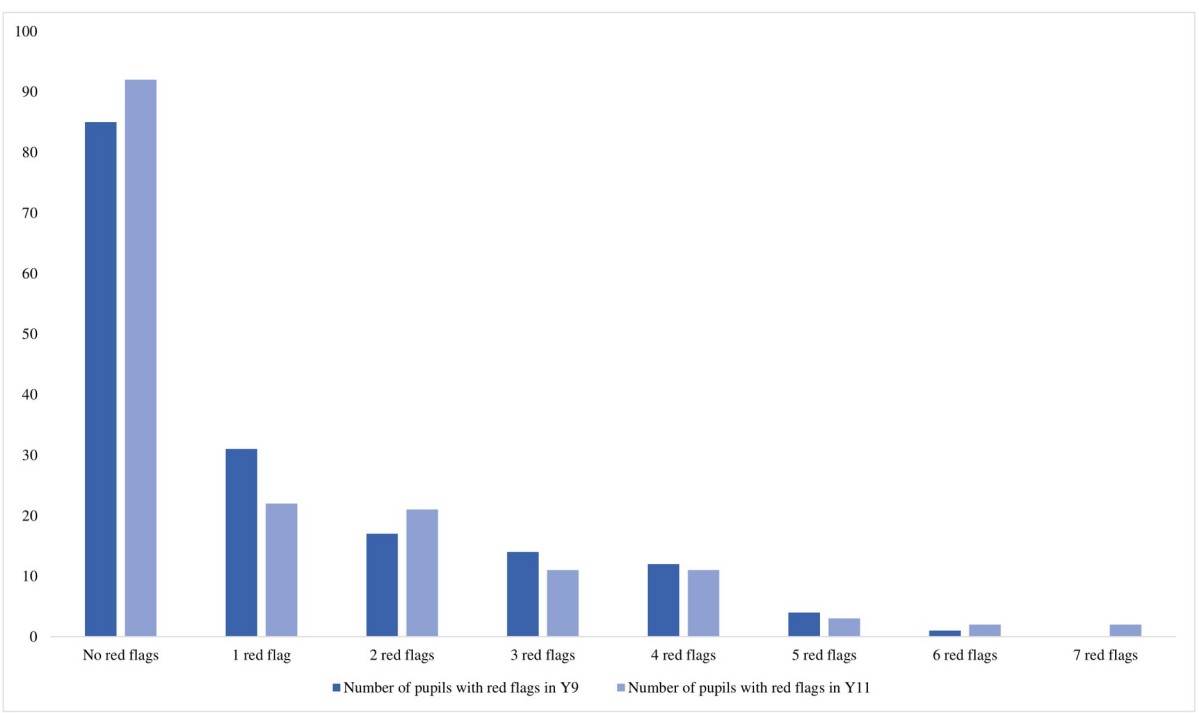

**Fig 1. Pupil red flag score in Year 9 and again in Year 11.**

## Principle component analysis

PCA resulted in four principle components with eigenvalues >1; two in Year 9 and two in Year 11. S4 Table presents the DHC screening tool questions grouped in each principle component. The principle components in Year 9 overlapped with the principle components in Year 11 but did not directly match. In Year 9, DHC questions on safety and online bullying grouped together, whilst in Year 11, DHC questions on concerns about body issues, sexuality and disabilities grouped together. Only principle component one was positively correlated with being offered a PHN(SN) referral appointment (Pearson's correlation = 0.53).

## Discussion

This is the first study to quantitatively evaluate the effectiveness of a locally commissioned, digital, health and wellbeing screening tool for schools in the UK, as part of a larger mixed-methods evaluation [16, 17]. Research calls for improved school-based interventions to prevent

**Table 3. Mean number of red flags in Year 11 and mean difference in number of red flags between Year 9 and Year 11 for three distinct groups of pupils.**

|  | N obs | N red flags in Year 11 | | Difference in N red flags from Year 9 to Year 11 | |
|---|---|---|---|---|---|
|  |  | *Mean* | *Range* | *Mean* | *Range* |
| Pupils with no red flags in Year 9 and did not attend Year 9 PHN(SN) appointment | 85 | 0.4 | 0–5 | 0.4 | 0–5 |
| Pupils with ≥1 red flags in Year 9 but did not attend Year 9 PHN(SN) appointment | 53 | 1.6 | 0–6 | -0.5 | -3–5 |
| Pupils with ≥1 red flags in Year 9 and did attend Year 9 PHN(SN) appointment | 24 | 2.7 | 0–8 | -0.3 | -4–4 |

Abbreviations: PHN(SN); Public Health Nurse (School Nursing)

**Table 4. Fully adjusted negative binomial difference-in-difference model exploring the effect of attending a PHN(SN) referral in Year 9 on red flag score in Year 11 (n = 164 pupils).**

| Outcome = red flag score | Coefficient | 95% CI | | P value |
|---|---|---|---|---|
| Attended PHN(SN) appointment in Year 9 x Time point | | | | |
| Attended—Year 11 | -0.36 | -0.97 | 0.24 | 0.242 |
| Attended PHN(SN) appointment in Year 9 | | | | |
| Attended | 0.26 | -0.19 | 0.72 | 0.249 |
| Time point | | | | |
| Year 11 | 0.14 | -0.18 | 0.46 | 0.385 |
| Number of red flags in Year 9 | 0.48 | 0.40 | 0.56 | <0.001 |
| Gender | | | | |
| Female | 0.27 | -0.06 | 0.59 | 0.111 |
| IMD | 0.02 | -0.03 | 0.06 | 0.472 |

Abbreviations: PHN(SN); Public Health Nurse (School Nursing), IMD; Index of Multiple Deprivation

Model AIC = 647.615

poor mental health and wellbeing among children and adolescents at a population-level [8]. Mental health screening in schools is suggested as a way to identify and prevent poor mental health but to date few schools in the UK have implemented screening tools [21]. Using a quasi-experimental approach, this study explored the effect of the DHC implemented in schools in the East Midlands of England on the number of pupils referred to Public Health Nurses (School Nurses) and whether identification of unmet needs in Year 9 resulted in a reduction in reported mental health issues in Year 11. Our findings indicate that the number of annual referrals made to PHN(SN) in schools administering the DHC remained comparable to those using the traditional referral approach only after adjustment for school-level covariates. No school reverted entirely to the DHC-based screening tool. These findings indicate that the DHC may increase the overall number of PHN(SN) appointments because referrals resulting from the DHC would likely be in addition to referrals resulting from the traditional approach. However, the DHC allows PHN(SN) to structure and plan their time, and can be used in tandem with the more reactive traditional pathway, without loss of efficiency. This supports our qualitative findings, which suggested that PHN(SN) found the workload associated with the DHC manageable because of its structured and planned nature, as well a meaningful part of their role [16]. In addition, the DHC was perceived as an acceptable and useful approach to identifying unmet health needs by adolescents and school stakeholders taking part [16, 17]. Previous research suggests that although teachers may be able to identify and refer pupils who externally express signs of poor mental health through behavioural problems, this is much more difficult for pupils whose mental health problems are more internalized (e.g., emotional problems) [22, 23]. Using digital screening tools such as the DHC, alongside the traditional pathway of teacher and pupil referral, may help identify pupils with either behavioural or emotional mental health issues, or both [16, 17].

When exploring number of annual referrals in schools with a lower compared to a higher percentage of pupils with English as their first language, our findings indicated that the association between participation in the DHC and increased number of annual PHN(SN) referrals was only observed in schools with a lower percentage of pupils with English as their first language. This aligns with evidence from elsewhere; a previous study indicated that children with English as an additional language had fewer behavioural mental health issues but more emotional issues (e.g., anxiety) [24]. Our qualitative findings suggest the DHC acted to increase

knowledge of and access to the resources and support provided by PHN(SN) [16, 17]. The current quantitative findings build on this and could be interpreted as indicated that the DHC raised awareness of where to seek help among pupils who do not have English as their first language or prompted teachers to refer pupils with emotional difficulties. However, further qualitative exploration would be needed to strengthen this conclusion.

Among the pupils who completed the DHC in Year 9 and again in Year 11, we observed a 4% decrease in the number of pupils with one or more red flags in Year 11 compared to in Year 9. However at the individual level, indicated by our DiD results, we did not find evidence of a reduced number of red flags in Year 11 among pupils attending a referral appointment in Year 9. One possible explanation might be changes in the health and wellbeing issues adolescents face over the school years, as our principle component analysis showed that DHC items grouped together slightly differently in Year 9 compared to Year 11. Therefore, pupils who attended their referral appointment in Year 9 may have resolved the issues facing them at the time but then faced new and different issues in Year 11. Another explanation might be that pupils who are referred do not receive effective follow-up care (provided by PHN(SN) and/or wider support services (e.g., CAMHS)). In a previous qualitative study, adolescents in England described a lack of mental health services, as well as long waiting lists to access those that were available (e.g., CAMHS). Adolescents also felt there was a lack of preventative provision (such as youth groups) in their local communities [25]. The qualitative evaluation [16, 17] corroborate the findings of our study in that the DHC helped to detect unmet health needs, promote awareness and encourage use of support options and inform the delivery of mental health support systems in schools. However, improvements to wider mental health support, which adolescents may be referred to by PHN(SN), in terms of funding, availability and accessibility may be required to improve and sustain population-level adolescent mental health.

## Study strengths and limitations

This is the first study using a quasi-experimental approach to evaluate a digital health and wellbeing screening tool implemented in schools in England. However, it is important to acknowledge the potential study limitations. Our analyses were conducted using a relatively small sample size (36 schools for school-level analysis and 164 pupils for pupil-level analysis). Furthermore, because of the quasi-experimental nature of the study there may have been differences between the DHC intervention schools and the control schools not accounted for in the analyses. Our outcome variable, annual number of PHN(SN) referrals did not include referrals resulting from the DHC, therefore the total number of referrals made through the traditional pathway plus DHC pathway is not known. In addition, the outcome variable represented the number of referrals across all school years. However, only Year 7, 9 and 11 pupils had the opportunity to complete the DHC and therefore any impact effect may have been diluted by the many pupils in other years who contribute to referrals but did not complete the DHC. Sixty-eight percent of intervention schools only took part in the DHC for one year and there was a lack of referral data previous to taking part in the DHC, limiting our analysis on trends over time. We cannot rule out control schools may have declined to take part in the DHC due to having alternative approaches to monitor pupil mental health. Finally, we did not have anxiety and depressive symptoms measured by validated assessment tools [26], but instead relied on the number of 'Red flags' as a proxy for pupil health and wellbeing. This is likely to have introduced measurement error and could have masked potential improvements.

It is also important to consider the impact of the COVID-19 pandemic on this study. Fewer schools took part in the DHC in the academic years 2019/20 and 2020/21 during the

pandemic, likely due to increased school pressures [16, 27]. This may have impacted our comparisons, due to having a smaller sample of intervention versus control schools during these years.

## Conclusion

Our findings suggest that the DHC did not impact referrals to the Public Health School Nurses and did not reduce the number of times pupils were raised as 'red flags' on the DHC survey from Year 9 to Year 11. However, our findings triangulated with the findings of our qualitative evaluation [16, 17], suggests digital screening tools such as the DHC may help to raise awareness of the mental health and wellbeing support in schools and detect unmet health needs. The DHC may be particularly helpful in schools where many pupils do not have English as their first language. However, wider health and wellbeing support, which takes into account how adolescents' issues can change over time may be required to achieve sustained improvement to health and wellbeing. Further co-production work with schools and pupils to improve the implementation and effectiveness of the DHC should be conducted before further roll-out of the programme to other cities in the UK.

## Supporting information

**S1 Fig. DHC flow chart.**
(DOCX)

**S1 Table. School-level characteristics.** School characteristics reported from 2018/19 government statistics. Little variation in school characteristics were observed across the three year data period.
(DOCX)

**S2 Table. Doubly robust negative binomial mixed-effects regression models exploring interactions between taking part in the DHC and school-level covariates on the annual number of PHN(SN) referrals (n observations = 102, n schools = 35).** Five separate doubly robust interaction models were conducted due to the limited sample size. Only interaction effects are presented. Main effects are not presented. All models were adjusted for year, number of pupils taking part in the DHC, ever taken part in DHC, % pupils eligible for FSM, % pupils with English as first language, % girls, % pupils with SEN and school IMD.
(DOCX)

**S3 Table. Change in number of red flags between Year 9 and Year 11 among the sample of 164 pupils.** *Green = N decreased (improved), Blue = N stayed the same, Red = N increased (worsened).
(DOCX)

**S4 Table. Principle components, showing correlations between DHC survey items, completed by pupils in Year 9 and again in Year 11 (n = 164).**
(DOCX)

## Acknowledgments

We thank the schools who provided data for our analyses and the pupils who took part in the DHC.

## Author Contributions

**Conceptualization:** Hannah Fairbrother, Clare Mills, Sarah Tebbett, Frank De Vocht.

**Data curation:** Alice Porter, Katrina d'Apice, Patricia Albers.

**Formal analysis:** Alice Porter.

**Funding acquisition:** Hannah Fairbrother, Frank De Vocht.

**Investigation:** Sarah Tebbett.

**Methodology:** Hannah Fairbrother, Frank De Vocht.

**Writing – original draft:** Alice Porter.

**Writing – review & editing:** Alice Porter, Katrina d'Apice, Patricia Albers, Nicholas Woodrow, Hannah Fairbrother, Katie Breheny, Clare Mills, Sarah Tebbett, Frank De Vocht.

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
