## [Decision Letter · Decision Letter 0]

6 Nov 2023

PONE-D-23-02654The impact of the newly developed school-based ‘Digital Health Contact’ – evaluating a health and wellbeing screening tool for adolescents in EnglandPLOS ONE

Dear Dr. Porter,

Thank you for submitting your manuscript to PLOS ONE. After careful consideration, we feel that it has merit but does not fully meet PLOS ONE’s publication criteria as it currently stands. Therefore, we invite you to submit a revised version of the manuscript that addresses the points raised during the review process.

Dear AuthorThank you for the paper sent. It has been very difficult to find reviewers to review your workPlease pay attention to this major comments. We would like to give the opportunity to review the while we find another strong reviewer for your workBest regards==============================

We look forward to receiving your revised manuscript.

Kind regards,

Juan A López-Rodríguez

Academic Editor

PLOS ONE

Journal Requirements:

2. Thank you for providing the following Funding Statement: 

“AP, KD, PA, NW, HF, KB and FDV have declared that no competing interests exists.

I have read the journal's policy and authors CM and ST have the following competing interests: CM is employed at the DHCs commissioning organisation (at Leicester City Council), and ST is employed at the DHCs provider organisation (at Leicestershire Partnership NHS Trust). This study, and the wider research project evaluating the DHC, has been co-produced with the DHC’s commissioners (at Leicester City Council) and providers (at Leicestershire Partnership NHS Trust). The commissioner and the provider are committed to helping to facilitate a robust evaluation of the DHC in order to inform potential changes and wider roll out of the programme. There has been no pressure or influence to modify, restate, weaken, omit or frame findings, conclusions and recommendations from any team member.”

We note that one or more of the authors is affiliated with the funding organization, indicating the funder may have had some role in the design, data collection, analysis or preparation of your manuscript for publication; in other words, the funder played an indirect role through the participation of the co-authors.

If the funding organization did not play a role in the study design, data collection and analysis, decision to publish, or preparation of the manuscript and only provided financial support in the form of authors' salaries and/or research materials, please review your statements relating to the author contributions, and ensure you have specifically and accurately indicated the role(s) that these authors had in your study in the Author Contributions section of the online submission form. Please make any necessary amendments directly within this section of the online submission form.  Please also update your Funding Statement to include the following statement: “The funder provided support in the form of salaries for authors [insert relevant initials], but did not have any additional role in the study design, data collection and analysis, decision to publish, or preparation of the manuscript. The specific roles of these authors are articulated in the ‘author contributions’ section.”

If the funding organization did have an additional role, please state and explain that role within your Funding Statement.

Please also provide an updated Competing Interests Statement declaring this commercial affiliation along with any other relevant declarations relating to employment, consultancy, patents, products in development, or marketed products, etc. 

Reviewers' comments:

Reviewer's Responses to Questions

**Comments to the Author**

1. Is the manuscript technically sound, and do the data support the conclusions?

Reviewer #1: Partly

Reviewer #2: Yes

2. Has the statistical analysis been performed appropriately and rigorously? 

Reviewer #1: Yes

Reviewer #2: Yes

3. Have the authors made all data underlying the findings in their manuscript fully available?

Reviewer #1: No

Reviewer #2: Yes

4. Is the manuscript presented in an intelligible fashion and written in standard English?

Reviewer #1: Yes

Reviewer #2: Yes

5. Review Comments to the Author

Reviewer #1: Dear Authors,

Thank you for submitting your work to the PLOS ONE journal.

The main goal of your work is to evaluate the effectiveness of a digital health and well-being screening tool, and you have called it Digital Health Contact, which is implemented in schools in the East Midlands of England…. Your manuscript needs improvement.

Here are some comments and suggestions:

Abstract:

- The manuscript does not have an abstract, making it difficult for the reader to understand what your manuscript will discuss.

Introduction:

- It’s too short and does not fully explain what your paper will discuss.

- I suggest you change the title of the introduction to Abstract, then after finishing the Abstract part (After discussion). You should write an introduction describing what your paper will discuss.

Background:

- The structure of the paper is not numbered and is very confusing.

Methods:

- It should contain an introductory paragraph to illustrate what this section will discuss.

Results

- It should contain an introductory paragraph to illustrate what this section will discuss.

Conclusion:

The future development of your work should be clarified more. Moreover, some achieved results should be mentioned in this part.

Thank you

Reviewer #2: The goal of this study was to evaluate the effectiveness of a digital health and wellbeing screening tool, which was implemented in several schools in East Midlands of England. The tool was used to identify pupils with unmet health needs and provide appropriate support.

The paper was well written and the analysis done in this work was clearly described.

6. PLOS authors have the option to publish the peer review history of their article (what does this mean?). If published, this will include your full peer review and any attached files.

Reviewer #1: **Yes: **Abdullah Almuhaimeed

Reviewer #2: No

---

## [Author Response · Author response to Decision Letter 0]

5 Dec 2023

Our responses to editor and reviewers' comments have been uploaded as a Word document in the document upload section as instructed in the email we received.

---

## [Editor Report · Decision Letter 1]

27 Dec 2023

The impact of the newly developed school-based ‘Digital Health Contact’ – evaluating a health and wellbeing screening tool for adolescents in England

PONE-D-23-02654R1

Dear Dr. Porter,

We’re pleased to inform you that your manuscript has been judged scientifically suitable for publication and will be formally accepted for publication once it meets all outstanding technical requirements.

Kind regards,

Juan A López-Rodríguez

Academic Editor

PLOS ONE
---

## [Editor Report · Acceptance letter]

4 Jan 2024

PONE-D-23-02654R1 

PLOS ONE

Dear Dr. Porter, 

I'm pleased to inform you that your manuscript has been deemed suitable for publication in PLOS ONE. Congratulations! Your manuscript is now being handed over to our production team.

Kind regards, 

on behalf of

Dr. Juan A López-Rodríguez 

Academic Editor

PLOS ONE